# Functional Near-Infrared Spectroscopy-Based Evidence of the Cerebral Oxygenation and Network Characteristics of Upper Limb Fatigue

**DOI:** 10.3390/bioengineering10101112

**Published:** 2023-09-22

**Authors:** Feng Li, Jiawei Bi, Zhiqiang Liang, Lu Li, Yu Liu, Lingyan Huang

**Affiliations:** Key Laboratory of Exercise and Health Sciences of Ministry of Education, Shanghai University of Sport, Shanghai 200438, China; lifeng_vanessa@outlook.com (F.L.);

**Keywords:** upper limb exercise, fatigue, functional connectivity, incremental load exercise, functional near-infrared spectroscopy

## Abstract

Objective: The objective of this research is to better understand the effects of upper limb fatigue on the cerebral cortex. The aim of this study was to investigate the characteristics of cerebral oxygenation and cortical functional connectivity in healthy adults after upper limb fatigue using functional near-infrared spectroscopy (fNIRS). Methods: Nineteen healthy adults participated in this study. The participants began exercising on an arm crank ergometer with no load, which was then increased by 0.2 kg per minute, maintaining a speed of at least 90 revolutions per minute during the exercise. Functional near-infrared spectroscopy covering the prefrontal cortex and motor area was used to monitor brain activity during rest and exercise. Heart rate and RPE were monitored during exercise to evaluate the degree of fatigue. Paired-sample *t*-tests were used to examine differences in the concentration of oxygenated hemoglobin (HbO_2_) and functional connectivity before and after fatigue. Results: All participants completed the exercise test that induced fatigue. We observed a significant decrease in HbO_2_ levels in the prefrontal and motor areas after exercise. In addition, brain network features showed a significant decrease in functional connectivity between the left and right motor cortices, between the motor and prefrontal cortices, and between both prefrontal cortices after fatigue. Conclusion: This study demonstrates that, in healthy adults, exercise-induced fatigue in the upper limbs significantly affects brain function. In particular, it leads to reduced functional connectivity between the motor cortex and the prefrontal cortex.

## 1. Introduction

Fatigue is a common phenomenon, especially exercise-induced fatigue [1]. Exercise-induced fatigue is generally defined as an increase in real or perceived difficulty and a decrease in performance associated with a task or exercise, where the muscles are unable to keep up with a specified level of intensity during exercise [2]. Fatigue is often caused by a combination of two factors, central and peripheral [3]. Central nervous system fatigue can be defined as a voluntary reduction in muscle activation, directly related to a reduction in motor neuron frequency and synchrony and a reduction in motor cortical drive [4]. Peripheral fatigue is defined as a reduction in the strength of muscle fiber contraction and alterations in the muscle action potential transmission mechanism [5]. When fatigue occurs, the peripheral system and the central nervous system can interact through metabolic and neurochemical pathways between skeletal muscle, the spinal cord, and the brain [6].

The topic of exercise-induced fatigue has been explored by many researchers, focusing more on whole-body and lower limb exercise-induced fatigue [4,7,8], and other studies have found that upper limb fatigue may have more profound effects than lower limb fatigue, leading to a more significant decrease in single leg standing balance [9,10,11]. In upper limb fatigue movements, the decrease in single leg standing balance may be partially due to central fatigue due to the absence of lower limb motor involvement, leading to a decrease in the ability to maintain functional tasks, such as basic support [10]. The “central control model” theory states that fatigue is the result of the entire central motor nervous control system actively modulating the neural control commands of motor units based on sensory afferent information from peripheral organs and the brain tissue itself [12]. While the majority of the current research has focused on the effects of upper limb exercise fatigue from the perspective of the peripheral factors [13,14,15,16,17], to the best of our knowledge, there are no studies exploring central changes after upper limb fatigue. Upper limb cycling training is mostly used in the field of rehabilitation and athletic training to function as aerobic or endurance training and cardiorespiratory testing and is an effective means of testing and developing upper limb exercise capacity [16]. In order to better exploit the benefits of upper limb training for rehabilitation populations or athletes, it is crucial to explore the brain mechanisms that follow the onset of upper limb fatigue.

Functional near-infrared spectroscopy (fNIRS) is a suitable technique for understanding cortical activity and function during exercise. This technology evaluates cerebral cortex activity by measuring cerebral oxygenation during neural activity [18]. fNIRS offers benefits including portability, low-cost modalities, and the capability for real-time monitoring in natural settings [19]. Its signals are more resistant to motion artifacts compared to electroencephalography [20]. Relative to functional magnetic resonance imaging, fNIRS has a higher temporal resolution and requires less space for data collection, providing critical information for interpreting brain responses related to motor performance [21]. fNIRS is commonly used in the motor field to monitor brain activity and functional connectivity [18,22,23]. Therefore, to better understand the effects of upper limb fatigue on the central nervous system, this study aimed to investigate cerebral oxygenation and cortical functional connectivity features following upper limb fatigue using fNIRS. We hypothesized that, after upper limb fatigue, both the cerebral oxygenation and the functional connectivity of the cortical regions would decrease.

## 2. Materials and Methods

### 2.1. Participants

A total of 19 healthy participants (11 females and 8 males; age, 22.7 ± 2.6 years; height, 169.50 ± 7.34 cm; weight, 59.52 ± 8.09 kg), recruited from the Shanghai University of Sport, participated in this study. None of the participants had injuries or illnesses that could affect their exercise performance. All participants play recreational exercise (jogging, cycling, etc.) one to three times a week, but no one trained regularly for any particular sporting event. Each participant was instructed to avoid vigorous exercise, stay up late, and consume caffeine or alcohol 24 h before the experiment. This study was approved by the Ethics Committee of Shanghai University of Sport (NO. 102772023RT031). All participants provided written informed consent after fully understanding the study procedures.

### 2.2. Procedures

Upon arrival at the laboratory, the participants were asked to rest for 5–10 min in a chair with a backrest. The seat position of the arm crank ergometer (Monark 891E, Stockholm, Sweden) was adjusted to ensure that it was aligned with the participant’s body midline, and that their arms were slightly bent when extended, with both feet flat on the floor [24]. Participants wore a fNIRS cap and heart rate (HR) monitor and remained seated in a resting state for 5 min. Subsequently, participants completed a fatigue exercise test using the arm crank ergometer. Borg’s rating of perceived exertion (RPE) (or 6−20 Borg) was used to monitor subjective fatigue levels at the end of each minute of the test. The 6–20 Borg scale ranges from “6” (no exertion) to “20” (maximal exertion) [25]. Meanwhile, a HR sensor (Polar H10, Polar Electroy, Kempele, Finland) was used to continuously monitor HR, which was recorded at the end of each minute.

We induced physical fatigue in the upper limbs using an incremental load exercise procedure described in the previous study [26]. Participants completed a 3 min warm-up on the ergometer and then started the exercise test with zero load as the first level, incrementing the weight load by 0.2 kg per minute. Throughout the exercise, the participants maintained a speed of 90 revolutions per minute (rpm) or higher while receiving verbal encouragement from the experimenters and pushing themselves until the end of the exercise test. The exercise concluded when participants met at least two of the following criteria: HR exceeding 180 beats per minute, RPE values reaching 18, or inability to maintain a speed of 90 rpm for a duration exceeding 5 s, even after receiving verbal cues and encouragement from the experimenters [24]. After the exercises, the participants still rested on a chair for 5 min.

### 2.3. Functional Near-Infrared Spectroscopy System Acquisition

This experiment adopts the multi-channel, continuous-wave desktop fNIRS system (NIRScout, NIRx Medical Technologies LLC, Glen Head, NY, USA) with a sampling frequency of 3.91 Hz, equipped with 16 detectors and 16 sources with dual wavelengths of 780 and 830 nm. The distance between the detectors and sources was set to 30 mm, producing 51 channels covering the prefrontal and motor cortices of the brain (Figure 1). To wear the fNIRS head cap, the experimenter located the Cz point, which is the intersection between the line connecting the nasion and the inion and the line connecting the external auditory meatus [27]. After the cap was fitted, the source and detector probes were fixed using a snap button in a plastic holder to ensure adequate contact with the participant’s scalp. The brain was scanned continuously for 5 min during the resting state before and after the exercise.

### 2.4. Data Pre-Processing

Data preprocessing of the fNIRS signals was performed using MATLAB (Version R2017a MathWorks, Natick, MA, USA) and the open-source package Homer 2.0 [28]. The preprocessing steps for the fNIRS data were as follows: (1) manual identification and marking of bad channels and artifacts; (2) conversion of the raw intensity-type signals to the optical density; (3) identification and removal of artifacts that were irrelevant to the experimental data: for each channel, data points with a fluctuation that exceeded three times the mean or 30 times the standard deviation within 1 s were marked as artifacts and removed within a 2 s window; (4) band-pass filtering, with a high-pass cutoff frequency of 0.01 Hz to remove instrument noise and low-frequency drift and a low-pass cutoff frequency of 0.1 Hz (third-order Butterworth filter) to eliminate physiological noise; and (5) conversion of optical density to changes in the hemoglobin concentration using the modified Beer–Lambert law to obtain the concentration of oxygenated hemoglobin (HbO_2_) and deoxygenated hemoglobin (HbR).

### 2.5. Functional Connectivity Analysis

We extracted the 3 min pre-exercise and post-exercise data for subsequent functional connectivity analysis. Pearson correlation analysis was conducted on the HbO_2_ from each time point for each participant to calculate the correlation coefficients between every pair of channels in the time series in order to assess the functional connectivity between the channels. The calculated indices were Fisher-Z transformed to ensure the normality of the data. The transformed numerical Z-score was defined as the work between the channels’ energy connection strength.

### 2.6. Statistical Analysis

The statistical analysis was implemented in MATLAB and SPSS version 27.0 (IBM Inc., Chicago, IL, USA). For all analyses, normality is checked before performing parametric tests (Shapiro–Wilk test). Data are reported as mean ± SD. Paired-sample *t*-tests were performed on the functional connectivity matrix before and after exercise fatigue to compare differences in functional connectivity strength; paired-sample *t*-tests were performed on the HbO_2_ values of each channel before and after exercise fatigue to compare differences in brain activity. To control for multiple comparisons, we applied a false discovery rate (FDR) correction to the resulting *p*-values, with a significance threshold set at an FDR-adjusted *p*-value < 0.05.

## 3. Results

### 3.1. Cerebral Oxygenation Changes

After exercise-induced fatigue, significant reductions in HbO_2_ levels were observed in the prefrontal and motor areas. Changes in the prefrontal cortex (PFC) occurred in the following areas: the orbitofrontal cortex (channels 1, 2, 4, and 7), frontopolar area (channels 3, 5, and 8), dorsolateral prefrontal cortex (channels 6, 9, 13, 14, and 17), ventrolateral prefrontal cortex (channels 18 and 32), and frontal eye fields (channels 16 and 23). Changes in the motor cortex occurred in the following areas: pre-motor and supplementary motor cortex (channels 26, 28, 39, and 41), primary motor cortex (M1) (channel 40), and primary somatosensory cortex (S1) (channels 46 and 50). Table 1 presents information on the significantly altered channels and the corresponding brain regions.

### 3.2. Functional Connectivity Changes

The analysis revealed changes in functional connectivity before and after fatigue in the prefrontal and motor areas, as well as between the left and right hemisphere motor areas and prefrontal regions. The functional connectivity changes between 51 channels before and after fatigue are depicted in a matrix representation, as illustrated in Figure 2. Channels that exhibited a significant decline post-fatigue are highlighted in Figure 3A. Furthermore, channels between the left and right hemispheres that showed a pronounced decrease after fatigue are presented in Figure 3B. Detailed information regarding the channels with significant differences, and their corresponding brain regions’ functional connectivity alterations, can be found in Table 2.

### 3.3. Physiological Response to Fatigue

As the load intensity increased, participants’ HR gradually increased. Fatigue was induced in the participants by increasing the load, with an average maximum HR of 180 beats/min. Additionally, the RPE increased with exercise intensity. On average, the participants reached the RPE of 19 when they were fatigued (Figure 4).

## 4. Discussion

This study focused on exploring the changes in cortical activity and network connectivity characteristics before and after upper limb fatigue. Our findings revealed a decreased activity in the prefrontal and motor cortices. Additionally, there was a reduction in the strength of functional connectivity in the brain network, particularly between the prefrontal and motor cortices, between both prefrontal cortices, and between both motor cortices. As hypothesized, our findings showed a significant decrease in cerebral oxygenation and functional connectivity in the brain after upper limb fatigue, which may impair communication and coordination among different brain regions. To the best of our knowledge, this is the first study to investigate the impact of upper limb fatigue on brain functional connectivity, providing insight into the influence of upper limb fatigue on the cerebral cortex.

During the exercise inducing upper limb fatigue, all participants reached a self-reported level of “very hard” or “exhausted”. The average HR for the majority of participants during fatigue stood at 157 beats/min. This aligns with prior research indicating a lower HR performance during upper limb exercises compared to whole-body or lower limb exercises [15]. This may be due to the lower proportion of type I muscle fibers in the upper limb muscle tissue (approximately 30%), compared to lower limb muscle tissue (approximately 50%) [29,30,31]. Lower muscle vascular conductance and maximal cardiac output in the upper limb muscles may have contributed to this effect [32]. Consistent with prior fatigue research where the most intense fatigue was observed at the end of exercise [33,34], our participants self-reported an RPE of “very hard” or “exhausted” after upper limb cycling, affirming the successful fatigue induction in this study.

Indeed, various studies have identified cortical regions associated with motor control in humans, such as the supplementary motor area, PFC, premotor cortex, M1, S1, and sensorimotor cortex [35,36,37,38]. According to the neurovascular coupling principle, in the resting state, a neuron has a constant fraction of oxygen extraction and a constant ratio of oxygenated to deoxygenated blood within the surrounding capillaries [39]. When moving from the resting state to the active state, there is a corresponding increase in HbO_2_ to meet the oxygen demand of the activity [40,41]. An increase in HbO_2_ can signify an enhanced neuronal excitability [40]. Herein, significant reductions in HbO_2_ concentrations were found in these relevant regions after fatiguing exercise of the upper limbs, indicating reduced neuronal activity after fatiguing exercise relative to the usual resting state.

Previous studies have shown that unilateral upper limb fatigue is followed by bilaterally reduced proprioception [42]. Athletes who perform 4 km of lower limb cycling after upper limb cycling reduced lower limb cycling performance [26], showing that upper limb fatigue can affect the performance of other body functions. Our findings provide a new possible explanation for this phenomenon, where reduced functional brain connectivity could better explain the negative effects of upper limb fatigue. We found, in agreement with previous studies, that, in addition to a reduction in the strength of functional connections between the bilateral motor cortical area hemispheres after a fatigue task [43], we also found a reduction between the frontal and motor cortices, between the bilateral prefrontal cortices, and between the prefrontal and contralateral motor cortices. This may be due to the fact that changes in motor commands associated with fatigue are regulated by the prefrontal lobe, which is responsible for action plans and rhythmic strategies, as well as decision-making [43]. The motor cortex and the ‘downstream’ efferent neural structures then directly control movement [44]. In movements of increasing intensity, fatigue affects not only the cortical responses related to the execution and feedback processing of information, but also the degree of resting state connectivity between different cortical areas [44]. Our study supports research on the effects of fatigue on brain network connectivity [45].

Studies have indicated that rhythmic upper limb cycling training may help older people improve their balance, improve walking function in stroke patients, and improve quality of life [46,47]. At higher exercise intensities, extensive cortical activation may increase information processing speed [48]. A certain intensity of exercise may improve connectivity between brain regions, increase the efficiency of information transfer, and improve motor performance and balance [49]. The present study found a significant reduction in functional brain connectivity following upper limb fatigue. This may be unfavorable for patients with conditions where the brain state is ‘fatigued’ (e.g., Alzheimer’s disease) [50] and they may need additional precautions to control exercise intensity to reduce the onset of fatigue.

In the current study, we have discussed differences in brain activity and cortical network connectivity before and after upper limb exercise fatigue. However, it is noteworthy that the following limitations remain. We have only used HR and RPE to measure fatigue and lack objective indicators to identify subdivisional fatigue-generating sites. Surface electromyography is a useful tool for studying changes in muscle neuromuscular activity as a diagnostic tool to measure and assess muscle contraction, which provides immediate data on neural activity within a specific muscle [51,52]. It has been mentioned that resting-state cortical connectivity can be used as an indicator of fatigue recovery [43]. The present study did not record the length of time that cortical activity and connectivity levels returned to pre-exercise, and this recovery due to exercise-induced fatigue will probably also be reflected in the degree of functional brain connectivity. 

In the future, we will attempt to use fNIRS to observe brain functional connectivity across various stages of exercise. The objective of this approach is to pinpoint the precise moments and conditions under which exercise fatigue manifests, as well as to understand the long-term implications of fatigue on the brain. Additionally, we aim to incorporate more comprehensive diagnostic tools, such as surface electromyography, to offer a holistic perspective on muscular and neural responses during physical exertion. By undertaking these measures, we aspire to devise more accurate and tailored exercise protocols, ensuring individuals reap the maximum benefits from their workouts. 

## 5. Conclusions

Our study indicated that cerebral oxygenation in the motor and prefrontal cortices decreased, and there was a notable reduction in the strength of functional connectivity between brain regions, following upper limb incremental load exercise-induced fatigue in healthy adults. The outcomes of this research contribute to a deeper comprehension of the mechanisms behind upper limb fatigue and its repercussions on brain function. This helps to deepen our understanding of the complex relationship between exercise-induced fatigue and central neural responses. And it also provides insights that can inform subsequent investigations in the fields of rehabilitation, athletic training, and daily human performance.

## Figures and Tables

**Figure 1 bioengineering-10-01112-f001:**
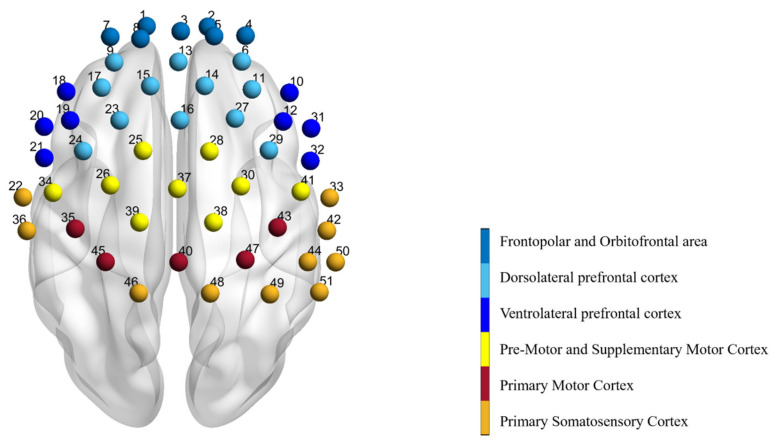
Graph of 51 measurement channels detected by fNIRS. The corresponding area of each channel is marked by a different color. The prefrontal lobe is divided into the frontopolar area, the orbitofrontal area, the dorsolateral prefrontal lobe, and the ventrolateral prefrontal lobe. The motor area includes the pre-motor cortex, supplementary motor cortex, and the primary motor cortex. Finally, there are the primary somatosensory cortex and the somatosensory association cortex.

**Figure 2 bioengineering-10-01112-f002:**
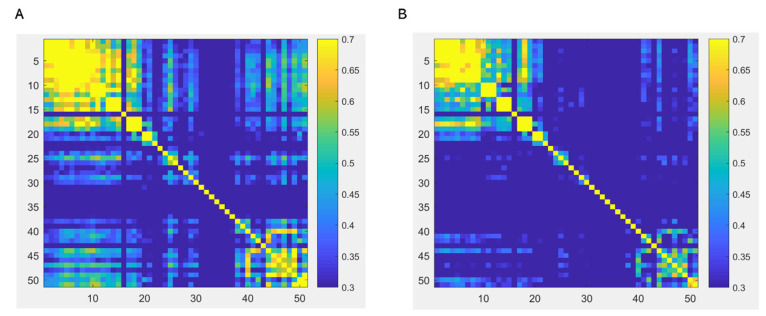
Functional connection strength matrix, which represents the correlation between channels. (**A**) represents the functional connectivity strength prior to fatigue, while (**B**) depicts the strength post-fatigue. Within the matrix, edges symbolize the channels, and nodes indicate the correlation amongst various channels. The color bar on the right signifies the Z-values of functional connectivity strength, with larger values denoting greater connection strength.

**Figure 3 bioengineering-10-01112-f003:**
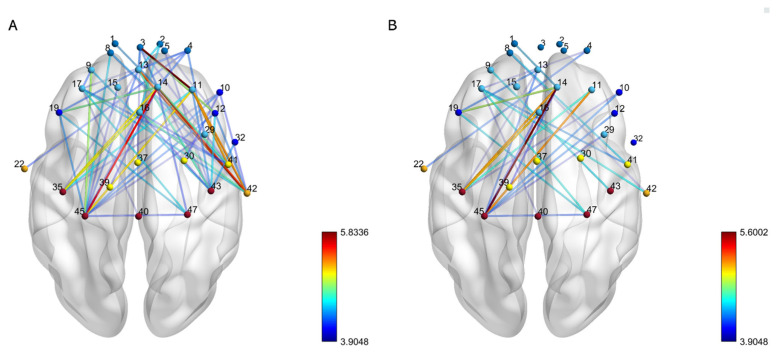
The functional connectivity visual map. The changes in the strength of functional connections between channels before and after fatigue. The connecting lines represent the differences in functional connectivity strength pre- and post-fatigue. The color bar indicates the T-values of the difference in functional connectivity strength between the two states. A redder hue on the line signifies a larger difference in functional connectivity post-fatigue compared to pre-fatigue, while a bluer hue indicates a smaller difference. (**A**) presents all channels with significant differences, while (**B**) highlights channels with pronounced differences between the left and right hemispheres.

**Figure 4 bioengineering-10-01112-f004:**
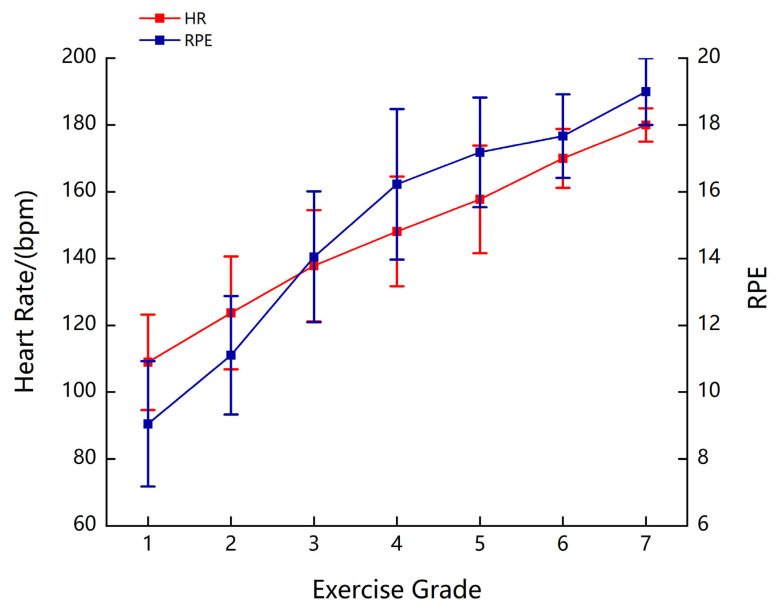
HR and RPE at each level of exercise.

**Table 1 bioengineering-10-01112-t001:** Channels and brain regions with reduced HbO_2_ levels after fatigue.

Area	Channel	Pre-Resting (×10^−4^ mmol/L)	Post-Resting (×10^−4^ mmol/L)	T	*p*
Orbitofrontal area	Ch1	0.53 ± 4.57	14.24 ± 12.41	−5.1650	<0.001
	Ch2	−0.46 ± 3.75	−14.22 ± 12.97	−4.9246	<0.001
	Ch4	0.17 ± 3.10	−5.87 ± 7.40	−3.1858	0.005
	Ch7	0.01 ± 3.40	−8.71 ± 10.83	−3.1562	0.006
Frontopolar area	Ch3	−0.08 ± 3.17	−6.46 ± 9.86	−2.5924	0.019
	Ch5	0.60 ± 3.21	−3.74 ± 3.48	−3.9272	0.001
	Ch8	0.10 ± 2.97	−6.62 ± 6.39	−3.9427	0.001
Dorsolateral prefrontal cortex	Ch6	0.16 ± 3.24	−3.81 ± 5.50	−2.6980	0.015
	Ch9	0.40 ± 1.92	−6.89 ± 7.04	−4.0708	0.001
	Ch13	0.56 ± 4.31	−6.05 ± 11.03	−2.4281	0.027
	Ch14	0.53 ± 3.35	−3.30 ± 5.65	−2.7408	0.014
	Ch17	3.16 ± 9.84	−7.24 ± 9.30	−2.4574	0.025
Ventrolateral prefrontal cortex	Ch18	2.13 ± 6.80	−3.43 ± 7.69	−2.9275	0.009
	Ch32	11.44 ± 37.49	−8.60 ± 29.87	−2.1332	0.048
Includes Frontal eye fields	Ch16	−0.41 ± 5.05	−7.16 ± 13.41	−2.1850	0.043
	Ch23	0.73 ± 3.91	−5.50 ± 9.19	−3.6556	0.002
Pre-Motor and Supplementary Motor Cortex	Ch26	2.08 ± 4.22	−4.52 ± 6.43	−3.7644	0.002
	Ch28	0.29 ± 8.68	−4.59 ± 7.64	−2.6413	0.017
	Ch39	1.28 ± 4.93	−3.42 ± 7.13	−2.1521	0.046
	Ch41	0.76 ± 5.94	−3.33 ± 6.66	−3.1625	0.006
Primary Motor Cortex	Ch40	0.56 ± 3.34	−2.07 ± 6.22	−2.3963	0.028
Primary Somatosensory Cortex	Ch46	1.01 ± 3.65	−1.98 ± 4.19	−2.2490	0.038
	Ch50	0.17 ± 4.11	−3.32 ± 4.03	−2.4860	0.024

Data are presented as mean ± SD.

**Table 2 bioengineering-10-01112-t002:** Areas of the significant reduction in functional joint strength before and after fatigue.

Region-Region		T-Value	*p*-Value
PM&SMA_L	VLPFC_L	−4.221	<0.001
PM&SMA_L	DLPFC_L	−4.27	<0.001
PM&SMA_L	DLPFC_R	−4.3057	<0.001
PM&SMA_L	VLPFC_R	−4.3882	<0.001
PM&SMA_L	VLPFC_R	−4.2824	<0.001
PM&SMA_L	SAC_R	−4.3782	<0.001
PM&SMA_L	DLPFC_M	−3.9482	<0.001
PM&SMA_R	OFA_L	−5.8336	<0.001
PM&SMA_R	FPA_L	−3.9229	<0.001
PM&SMA_R	FPA_L	−4.9266	<0.001
PM&SMA_R	DLPFC_L	−4.549	<0.001
PM&SMA_R	DLPFC_L	−4.3457	<0.001
PM&SMA_R	S1_L	−4.0253	<0.001
PM&SMA_R	DLPFC_R	−4.1395	<0.001
PM&SMA_R	VLPFC_R	−4.2891	<0.001
PM&SMA_R	VLPFC_R	−4.5104	<0.001
PM&SMA_R	S1_R	−3.9672	<0.001
M1_L	DLPFC_M	−3.9406	<0.001
M1_R	OFA_L	−5.3158	<0.001
M1_R	FPA_L	−4.3207	<0.001
M1_R	DLPFC_L	−5.3984	<0.001
M1_R	OFA_R	−3.9362	<0.001
M1_R	DLPFC_R	−4.1346	<0.001
M1_R	VLPFC_R	−5.4862	<0.001
M1_R	VLPFC_R	−4.7553	<0.001
M1_R	DLPFC_M	−4.524	<0.001
M1_M	OFA_L	−4.5653	<0.001
M1_M	DLPFC_L	−3.9048	<0.001
M1_M	DLPFC_R	−3.9669	<0.001
M1_M	VLPFC_R	−5.1262	<0.001
M1_M	VLPFC_R	−5.0606	<0.001
DLPFC_L	DLPFC_L	−4.267	<0.001
DLPFC_L	VLPFC_R	−4.8204	<0.001
DLPFC_M	FPA_R	−4.2324	<0.001

L, left; R, right; DLPFC, Dorsolateral prefrontal cortex; VLPFC, Ventrolateral prefrontal cortex; OFA, Orbitofrontal area; FPA, Frontopolar area; PM&SMA, Pre-Motor and Supplementary Motor Cortex; M1, Primary Motor Cortex; SAC, Somatosensory Association Cortex; S1, Primary Somatosensory Cortex. T indicates the difference in strength between before and after. A negative T value means post-fatigue < pre-fatigue.

## Data Availability

Not applicable.

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
