# Peer review of "Functional Near-Infrared Spectroscopy-Based Evidence of the Cerebral Oxygenation and Network Characteristics of Upper Limb Fatigue"

_bioengineering, 2023, doi:10.3390/bioengineering10101112_

Round 1

Reviewer 1 Report

In this study in humans, the authors are trying to correlate changes in intercortical activities measured by use of functional near-infrared spectroscopy and upper limb fatigue associated with intensive physical exercises. The material is described successively and conclusions are partially supported by obtained data.

Remarks/recommendations:

1)     the title is too ambitious as “the neural activity” was not measured directly;

2)     in the Abstract:

a) “…greater than 18…” should be either removed or clarified;

b) “All participants completed the exercise test…”;

c) the final sentence (“After upper limb…”) should be rewritten;

3)     in the Introduction:

a)    the sentence of “The causes of fatigue…” should be rewritten;

b)    a space should be inserted in “…drive [2].” and everywhere below;

c)    “The topic of exercise-induced fatigue…;

d)    “(EEG)” should be omitted as it was not used below;

4)     In the Materials and Methods:

a)    “(Monark 891E)” and “(Polar HR monitors)” need additional information;

b)    “…reminder stimulation…” should be clarified;

5)     In the Results:

a)    in Table 1, the title should be rewritten (“concentration” should be clarified);

b)    in Table 1, “Pre-resting” and “Post-resting” columns should contain the units;

c)    in the section of “3.2. Functional connectivity change”, the text and the legends to figures should be clarified and completely rewritten;

d)    “…reached the RPE of 19…”;

6)     in the Discussion:

a)    “…lower HR…”;

b)    “…may have contributed to this effect”. needs a reference;

c)    the last sentence on page 8 (“However, based…”) should be rewritten;

d)    the sentence of: “The HbO2 can represent the excitability (activity) of a neuron[30]” should be either clarified (HbO2 increase or its decrease) and rewritten or removed;

e)    “…bilaterally reduced proprioception [31].”;

f)     “…the bilateral motor cortical areas after a fatigue task [32].”;

g)    “Our study supports a research ….” needs a reference;

h)    “(EMG)” should be removed from the text.

7)     in the Conclusion:

a)    “…that reduced neural activity in…” should be replaced by “that activities in…”;

b)    “… exercise-induced…”.

English should be double-checked.

Author Response

We sincerely appreciate the valuable suggestions provided by the reviewers. We have carefully considered all of your comments and have made thorough revisions to the manuscript accordingly. Please refer to the attached point-by-point response letter for further details.

Reviewer 2 Report

1.       You are advised not to cite several times in turn the same reference, e.g. [2].

2.       “few studies have examined the central changes following upper limb fatigue.”. Name these studies.

3.       “spectroscopy (fNIRS) is a new technique for understanding cortical activity and function.”. I do not think that spectroscopy (fNIRS) is a new technique for understanding cortical activity and function. Can you prove your statement?

4.       fNIRS has advantages” in comparison with?

5.       “but no one trained regularly for any particular sporting event[12]”. Why do you cite [12]? What is a relation of the participants to [12]?

6.       HbO2 levels” – typo.

7.       “Indeed, various studies have identified cortical regions associated with motor control in humans, such as the supplementary motor area, PFC, premotor cortex, M1, S1, and sensorimotor cortex [28]”. If you write plural “studies”, you have to cite several references. Such a note is valid for several statements having “studies”.

8.       Conclusion section must provide more information of the research performed.

9.       “Our study revealed that reduced neural activity in the motor and prefrontal cortices, as well as a general reduction in the strength of functional connectivity between brain regions, were observed in healthy adults after undergoing upper limb incremental load exercise induced fatigue.”. We could guess the reduction of neural activity without your experiments. Have you observed some novel result in your experiments?

Some typos were noticed.

Author Response

(The authors gave the same response as above.)

Reviewer 3 Report

This is an interesting and well written manuscript. However, I have some comments and suggestions which may improve the quality of this paper.

The authors used a variety of fatigue terminologies. Central fatigue, peripheral fatigue, motor fatigue, perceived fatigue. Please be consistent.

Also, it should be clear that there is no single cause of fatigue. Always a combination of central and peripheral factors.

Are there any sex differences in the study?

Reduce the number of decimals of the p value.

What was the main criteria of task termination? Just RPE? Maybe the participants were not motivated enough?

This reviewer thinks that fatigue already started very early on of the task. The design is investigating the mechanisms of "task failure" not fatigue. Please comment. I suggest to review the fatigue literature, for example the work of Enoka et al. and Rudroff et al. about perceived fatigue, performance fatigability, and mechanisms of task failure.

Limitations are addressed fine. More information about potential future studies are needed.

Author Response

(The authors gave the same response as above.)

Round 2

Reviewer 1 Report

bioengineering-2621003: “Functional near-infrared spectroscopy-based evidence of the neural activity and network characteristics of upper limb fatigue”.

The authors have made a careful revision and responded to almost all points I raised.

Minor editing of English language required